# Double-Layer MWCNTs@HPPS Photothermal Paper for Water Purification with Strong Acid-Alkali Corrosion Resistance

**DOI:** 10.3390/membranes12121208

**Published:** 2022-11-29

**Authors:** Yuan Gao, Guoqing Jin, Shuaishuai Wang, Lihua Lyu, Chunyan Wei, Xinghai Zhou

**Affiliations:** 1School of Textile and Materials Engineering, Dalian Polytechnic University, Dalian 116034, China; 2Shandong Chambroad Holding Group Co., Ltd., Binzhou 256500, China

**Keywords:** multiwalled carbon nanotubes, PPS paper, solar driven evaporation, desalination

## Abstract

Solar-driven interfacial evaporation technology has been identified as a promising method to relieve the global water crisis, and it is particularly important to design an ideal structure of the solar thermal conversion evaporation device. In this paper, hydrophilic polyphenylene sulfide (HPPS) paper with loose structure and appropriate water transmission performance was designed as the based-material, and multi-walled carbon nanotubes (MWCNTs) layer with excellent photothermal conversion performance was constructed to realize the high-efficiency solar-driven evaporation. Under tail swabbing mode, the cold evaporation surface on the back of the evaporator greatly improved the evaporation rate, cut off the heat transfer channel to bulk water, and achieved the maximum evaporation rate of 1.23 L/m^2^·h. Ethyl cellulose (EC) was introduced to adjust the water supply performance of HPPS layer, and a large specific surface area of cold evaporation was obtained, thus improving the water evaporation rate. In the simulation experiment of seawater desalination and dye wastewater treatment, it showed good water purification capacity and acid/alkali-resistance, which had great practical application significance.

## 1. Introduction

The serious shortage of global freshwater resources for human life and industrial production has become a continuous hot topic, and with the wastewater discharge from the pharmaceutical, textile and metallurgical industries, this situation is intensifying [1,2]. The sea water stock on the earth is very large, accounting for 96.53% of the global water resources [3], developing desalination technology to obtain freshwater and solving water pollution can alleviate this dilemma [4,5]. Traditional water treatment methods directly or indirectly consume fossil energy in varying degrees, which will cause environmental pollution and waste of resources [6,7]. As a renewable energy, solar energy can be used to generate steam for desalination and wastewater purification, providing another sustainable solution to solve the problem of freshwater scarcity [8,9,10]. 

Human beings have a long history of using solar energy to drive evaporation [11], mainly in three forms: water bottom heating, overall heating, and surface heating [12]. The first two have the disadvantage of great heat loss, because water is a poor light absorber with low light heat conversion efficiency. When the incident light passes through the water surface and downward through the water body, the intensity will significantly decrease. In addition, the generated heat energy needs to heat the whole water body to produce steam. This process will have great heat loss, resulting in low evaporation efficiency [13]. Through the above analysis of water evaporation process, light absorption and heat management are important factors to improve light heat conversion efficiency and water evaporation rate. Researchers such as Halas et al. [14], Deng et al. [15], and Yuan et al. [16] used heavy metal nanoparticles as photothermal converters to drive the generation of water vapor, but the high cost of raw materials limits its wide application. Chen et al. [8] used cheap graphite powder and porous carbon composites to obtain solar energy through thermal localization to achieve more effective steam generation, and showing a solar thermal performance of up to 85%, and proposed a solar-driven evaporation material, which has the characteristics of absorbing solar spectrum, heat insulation, hydrophilicity, and interconnected pores. 

The purpose of heat management in solar-driven evaporation process is to suppress heat loss, including heat radiation, convection, and conduction [17]. The evaporation materials are always in direct contact with bulk water, resulting in heat loss (heat conduction) [14,18,19]. Therefore, it is a good idea to design a double-layer structure interface evaporation material with thermal insulation (low thermal conductivity layer) [20,21], and the system includes a high-efficiency photothermal conversion layer and a thermal insulation layer with capillary water absorption. The working process is as follows: the photothermal conversion layer converts the absorbed solar energy into heat energy and exists on the surface, then the water absorption layer absorbs water from the bulk water to the interface between the two layers via capillarity, which is heated and evaporated [20,22,23,24,25]. This process has been widely used in desalination by researchers, and solar-driven interfacial evaporation devices with various materials or structures have been developed. In the whole process, the photothermal conversion layer is far away from the bulk water to limit the heat conduction loss. 

In the desalination process, salt accumulation caused by water evaporation will lead to the continuous reduction of evaporation rate [26]. Multiple strategies are available to solve this problem, including mechanical removal, shielding effect, and force-driven fluid flow [26,27]. For example, capillary water absorption of the designed water supply channel is required to not only meet the demand of evaporation rate, but also have a small amount of circulating water to facilitate the diffusion of high concentration brine back to bulk water. In addition, taking advantage of the unique structure of Janus, two functions of steam generation, solar absorption, and water pumping, are decoupled into different layers, which can also effectively prevent salt from accumulating in the photothermal layer [28,29].

In this study, we used the “tail swabbing mode” evaporation device, and prepared loose and super-hydrophilic HPPS paper with suitable water-carrying capacity as the support layer of the photothermal conversion device. Adjust the water absorption structure to make the water delivery speed meet the interface evaporation, and there is also a regional water cycle between the evaporation interface and the water delivery layer, which promotes the salt to be transported to the bottom surface by gravity. At the same time, the water with heat on the top surface is transmitted to the loose layer, which will rely on its large specific surface area to realize the rapid increase of evaporation surface area and enhance the water evaporation rate.

## 2. Materials and Methods

### 2.1. Materials

Polyphenylene sulfide (PPS) powder was purchased from Ticona (Fortron 0320), and PPS chopped fiber was provided by Jinlun New Material Technology Co., Ltd., Tianjin, China. Benzophenone (BP), dibutyl phthalate (DBP), sodium dodecyl sulfate (SDS), and EC were obtained from Aladdin. HNO_3_ (66–67%) and HCl (37%) were provided by Fengchuan Chemical reagent Co., Ltd., Tianjin, China. MWCNTs (inner diameter: 3–5 nm, outer diameter: 8–15 nm, length: ~50 μm) was provided by Macklin Biochemical Technology Co., Ltd., Shanghai, China. NaOH, NaCl, KCl, CaCl_2_, and MgSO_4_ were purchased from Kermel Chemical Reagents Development Center, Tianjin, China. Methyl orange (MO) and Methylene blue (MB) were purchased from Sinopharm Chemical Reagent Co., Ltd., Shanghai, China. All materials were used without further purification.

### 2.2. Preparation of PPS Paper

Loose PPS paper was prepared by paper-making technology and hot-pressing technology. In this process, PPS solid solution powders prepared by thermally induced phase separation (TIPS) method was used as adhesive to bond PPS chopped fiber to obtain PPS paper with strong mechanical properties, and the preparation method of PPS solid solution powders was described in detail in the supporting information [30,31,32].

The specific operations were as follows: 4 g PPS chopped fiber and PPS solid solution powder with weight ratio of 1:2/1:3/1:4 (PPS fiber: powder) were uniformly dispersed in 0.8 g/L PEO aqueous solution in a high-speed disperser. The wet filter cake was obtained through paper-making operation, and then dried by blowing at 60 °C. Finally, the dried PPS filter cake was hot-pressed at 220 °C and 18 MPa for 3 min, soaked in ethanol to remove the solvent, and dried to obtain PPS paper. 

### 2.3. Preparation of MWCNTs@HPPS Paper

The simply nitric acid treatment process was adopted for hydrophilic modification of PPS paper [33,34]. First, PPS paper was soaked in 33–34% nitric acid aqueous solution, and the solution was heated to 60 °C. After 10 min, 20 min, or 30 min, the paper was washed to neutral for standby, named HPPS paper. MWCNTs (12 mg) and SDS (12 mg) were dispersed in ethanol/water (4/1, *v*/*v*) solution containing EC via intense ultrasound for 30 min. Subsequently, MWCNTs were deposited on the surface of HPPS paper by static gravity deposition to obtain MWCNTs@HPPS paper, and dried it for further testing. The illustration of the preparation process is shown in Figure 1.

### 2.4. Characterization

The surface morphology of as-prepared papers was explored via field emission scanning electron microscope (SEM, JSM-7800F, Tokyo, Japan). Fourier transform infrared (FTIR) spectroscopy spectra of the membranes were characterized using spectrometer (Thermo Fisher Nicolet FTIR, Waltham, MA, USA). X-ray photoelectron spectroscopy (XPS, Thermo Fisher K-alpha, Waltham, MA, USA) was conducted to explore the bonding environment and chemical composition, respectively. The static pure-water contact angle (WCA) was explored via contact angle measurement (JC2000D5H, POWEREACH, Shanghai, China).

### 2.5. Solar-Driven Water Evaporation Performance Measurement

The steam generation experiments were performed via a solar simulator (CEL-HXF300, CEAULIGHT, Beijing, China), and a standard AM 1.5G spectral filter was fixed in the simulator. Before each test, the light intensity must be calibrated with an optical power meter. The samples were cut into a disc with a diameter of 2.5 cm and placed in a 25 mL beaker containing deionized water. The samples could be tested by floating or tail insertion. The water mass change with irradiation time was recorded by electronic balance (accuracy 0.1 mg), and the real-time measurement was carried out to calculate the evaporation rate [35,36]. 

### 2.6. Solar-Driven Seawater Desalination and Wastewater Purification Experiments

NaCl, KCl, CaCl_2_, and MgSO_4_ were used to simulate seawater component, and NaCl aqueous solution with different concentrations were used to simulate the various seawater in the world. Two kinds of dyes, MO and MB, with the concentration at 20 mg/L were used to simulate the organic pollutants in wastewater. The salt concentration was tested by the conductivity meter (Rex Instruments Factory, DDS-307, Shanghai, China) and the absorption spectra of the dye aqueous solution was obtained using an ultraviolet-visible spectrophotometer (SolidSpec-3700, Kyoto, Japan) with absorptions of 300–1200 nm.

## 3. Results and Discussion

### 3.1. Structure and Performance Analysis of PPS, HPPS, and MWCNTs@HPPS Papers

In the process of making PPS paper, chopped fiber was the main body, and the solid solution powder played the role of linking fibers, which could also be called adhesive agent. This was because the powder contained solvent and porous PPS resin. Under the action of high temperature and high pressure, after the solvent in solid solution powder melted, the contacted fiber surface was micro-dissolved and bonded with the porous PPS resin, forming PPS fiber paper. As shown in Figure 2, with the increase of solid solution powder content in papermaking components, there were more connection points in the formed paper, which increased the tensile strength, even more than 15 MPa (1:4). As shown in Figure 3a–c, with the increase of the proportion of powder, the structure observed by electron microscope gradually became dense and the degree of porosity decreased. The solid solution powder was prepared by thermally induced phase separation (TIPS) method, and the PPS resin structure after removing the solvent showed a branch-like porous structure, which is shown in Appendix A. Further enlarged observation of the internal powder structure of prepared PPS paper, shown in Figure 3d–f, showed that it still maintained the branch-like porous structure, which was the similar to the powder structure before paper-making, and was not subjected to pressure causing hole extrusion deformation. After hydrophilic modification with nitric acid aqueous solution, the main structure and micro-morphology of the paper were not damaged. Only the PPS molecules in the outermost layer of the resin were modified to hydrophilic state, which is confirmed in our previous work [33]. After all, nitric acid, as a strong oxidizing acid, causes the partial chain break of PPS molecular chain [37], especially the impact on porous powder at the connection point was obvious, resulting in the decrease of the tensile strength of HPPS paper, but it could still be maintained above 11 MPa (1:4). As shown in Figure 3g, MWCNTs were evenly fixed on the surface by EC molecules, in sharp contrast to the loose surface of paper.

The chemical composition of PPS and HPPS papers were displayed by FTIR spectrum. As shown in Figure 4a, a new peak appeared at 1039.96 cm^−1^ belonging to the -SO- bond, which was due to the oxidation of -S- to -SO- by nitric acid [38]. The peaks emerging at 1641 and 738 cm^−1^ and became more intensive corresponding to bending vibration of N-H bonds of primary amine. The envelope-like peak at 3200–3700 cm^−1^ attributed to primary amines. As shown in Figure 4b, it could be seen from the XPS survey spectra that a small amount of N element (2.69%) appeared obviously, and the O/S ratio increased significantly (from 0.25 to 0.45), which proved that more oxygen-containing functional groups were introduced. N 1s and S 2p high-resolution spectra are shown in Figure 4c,d, the peaks at 406 eV, 399.8 eV, and 400.6 eV were the binding energy of -NO_2_, -C-N-, and -NH_2_ bonds, and the peaks at 163 eV and 164.2 eV belonged to S 2p3/2 and S 2p1/2, which were in agreement with the C-S bond [38], and the peak at 165.4 eV corresponds to -SO-, and the above conclusions were consistent with the FT-IR spectrum analysis.

As shown in Figure 5a–d, the color of the paper after hydrophilic modification with nitric acid obviously turned yellow, which was due to the color presented by the introduction of -NO_2_. The WCA test showed that the modification effect was remarkable, and the WCA changed from more than 85° of PPS paper to 0° of HPPS paper. After depositing carbon tubes on the top surface, showing black, while the bottom surface still showed yellow, which proved that MWCNTs only existed on the top surface, so as to realize the preparation of double-layer functional materials, including MWCNTs as photothermal conversion layer and hydrophilic paper as water transfer layer. Figure 5e showed the effect of the modification time of PPS paper (1:3) in nitric acid aqueous solution on the water drawing capacity. It was found that the wicking heights increased with the increase of modification time. When the modification time was 30 min, the wicking height within 150 s reached 7.9 cm, which far exceeded the water transmission capacity required for evaporation. Due to the superhydrophobic characteristics of MWCNTs, the top surface WCA of MWCNTs@HPPS paper was greater than 130°, which was conducive to the rapid evaporation of water and no water droplets gather on the surface, which was conducive to the stability of photothermal conversion efficiency. In addition, the hydrophobicity of the upper surface could also promote self-floating on the water surface, as shown in Figure 5f.

### 3.2. Light–Thermal Performance Analysis of MWCNTs@HPPS Paper

Photothermal conversion materials are very important for the performance of solar-driven interface evaporation devices. MWCNTs have been proved to have high light absorption efficiency (greater than 93%) [23,39,40]. Thus, MWCNTs were used to construct photothermal conversion layers in this work. As shown in Figure 6, compared with the pristine PPS paper, the prepared MWCNTs@HPPS paper showed high light-absorption. As shown in Figure 7a, the photothermal conversion performance of the prepared papers were investigated, under 1 sun, the surface temperature of MWCNTs@HPPS paper rapidly increased to more than 50 °C, then gradually increased to more than 60 °C, and reached the maximum surface temperature of 63 °C after 10 min. For the samples without MWCNTs layer, the surface temperature of PPS and HPPS papers had a relatively smaller increase (about 36 °C and 42 °C, respectively). After turning off the light, the surface temperature of prepared papers returned to room temperature soon. IR images showed the surface temperature, which intuitively showed the comparison of the surface temperature of the prepared papers. As shown in Figure 7b, the IR images at different time periods were captured simultaneously to carefully record the surface temperature changes of different samples (MWCNTs@HPPS papers under floating mode and tail swabbing mode), and the temperature changes curve within 10 min are shown in Figure 7c. The temperature distribution of bulk water under the two evaporation modes was obviously different, in the floating mode, there was obviously strong heat diffusion to bulk water, which was mainly due to poor insulation performance caused by the loose characteristics of PPS paper. In the tail swabbing mode, the MWCNTs@HPPS paper did not directly contact bulk water, which greatly limited the heat diffusion, so the temperature change of bulk water was weak and the paper surface temperature was higher than that used in the floating mode. The temperature curve along the axial direction of experimental modes under 1 sun after 10 min is shown in Figure 7d, which further confirmed the above conclusion.

### 3.3. Solar-Driven Water Evaporation Performance Analysis of MWCNTs@HPPS Paper

The solar-driven water evaporation performance of the prepared papers under different modes was investigated in our experiment, and was evaluated by monitoring the mass loss of water in beakers with time under 1 sun illumination. As shown in Figure 8a, Xenon lamp was used to simulate sunlight, and the electric balance was used to characterize the change of water quality during evaporation under sunlight irradiation time. Two different evaporation models, including floating mode and tail swabbing mode, were designed to test the evaporation rate. The cumulative water loss mass of different samples shown in Figure 8b increased linearly with illumination time. It was obvious that under 1 sun, the mass change of pure-water without evaporator reached 0.12 kg/m^2^ after 40 min (~0.18 kg/m^2^·h). For the prepared paper tested in floating mode, the mass change after 40 min was 0.42 kg/m^2^ (~0.63 kg/m^2^·h, evaporation efficiency at 44.68%), which was basically not affected by the paper composition. That was because the loose structure of paper was conducive to the circulation of water between the upper and lower layers of paper, resulting in heat loss. However, for the tail swabbing mode, the maximum mass change after 40 min was 0.82 kg/m^2^ (~1.23 kg/m^2^·h, evaporation efficiency at 87.23%), which was 95% higher than that under floating mode. There were two evaporation surfaces in the tail swabbing mode, including solar evaporation surface and cold evaporation surface, and the schematic diagram of evaporation device is shown in Figure 8c. The solar evaporation surface was the illuminated side, while the cold evaporation surface was located on the backlight side. The energy generated by the photothermal conversion layer, in addition to the energy required for water evaporation, heat radiation loss, and heat convection loss, the remaining energy was transmitted to the cold evaporation surface through heat conduction for water evaporation [41,42]. Water could circulate freely in loose HPPS paper, which facilitated heat conduction, and the body possessed a high specific surface area, resulting in a large area of cold evaporation surface.

In this work, EC not only acted as a binder to improve the interfacial adhesion between photothermal conversion layer and HPPS paper, but also could adjust the water transmission capacity of HPPs paper. As shown in Figure 9a,b, the addition of EC obviously affected the water transmission performance and hydroscopic rate of HPPS. Compared with HPPS paper, EC@HPPS paper prepared with 0.2 g/100 mL EC solution showed lower water absorption rate, and the hydroscopic rate decreased from 78.77% of HPPS paper to 53.18% of EC@HPPS paper, which proved that more pores were generate under saturated state. As shown in Figure 9c, EC component affected water mass changes, the appropriate amount of EC regulated the cold evaporation surface area under saturation, the prepared paper with 0.2 g/100 mL EC solution showed high water evaporation rate at 1.23 kg/m^2^·h and schematic diagram of water trace distribution on cold evaporation surface is shown in Figure 9d, while excessive EC would cause low water supply rate, resulting in low water evaporation rate. In addition, the prepared paper with 0.2 g/100 mL EC solution exhibited excellent long-term evaporation stability for 12 h, where the evaporation rate remained above 1.10 kg·m^−2^·h^−1^ (Appendix A).

### 3.4. Seawater Desalination and Wastewater Purification Performance Analysis of MWCNTs@HPPS Paper

As shown in Figure 10a, four kinds of salts, NaCl, KCl, CaCl_2_, and MgSO_4_, were used to simulate the salt components in seawater. After desalination by MWCNTs@HPPS paper, the salt concentration of the collected condensed water was greatly reduced, which was in line with the guidelines of the WHO drinking water standards [43,44,45]. As shown in Figure 10b, several representative seawater samples with different NaCl concentrations at 0.8 wt% (Baltic Sea), 3.5 wt% (World Ocean), 4.1 wt% (Red Sea), and 10 wt% (Dead Sea) were utilized for solar seawater desalination experiments. After desalination by MWCNTs@HPPS paper, the salinity of the collected condensed water decreased significantly, far lower than the guidelines of the WHO drinking water standards, showing excellent seawater desalination capacity. In addition, dye wastewater and acid-base wastewater with 1 M HCl or 1 M NaOH were also used to test the wastewater treatment capacity of paper, shown in Figure 10c,d. After purification, the color of methyl orange and methylene blue aqueous solution changed from colored solution to colorless solution, and the UV-Vis spectrum showed that the characteristic peak disappeared and the absorbance decreased to 0. The acid-base wastewater before purification was detected by pH test paper and showed strong acid and alkali, while the purified liquid detected by pH test paper was the same as that of deionized water, which proved that it possessed strong corrosive sewage treatment capacity.

## 4. Conclusions

In conclusion, based on mature papermaking technology and hot-pressing process, we have developed a double-layer MWCNTs@HPPS Janus paper for solar-driven interface evaporation. Especially under the tail swabbing mode, the water evaporation rate of the prepared paper reached 1.23 L/m^2^·h, which was mainly due to interfacial evaporation and cold surface evaporation. By adjusting the water absorption of the cold evaporation surface, the evaporation area of the cold evaporation surface could be significantly increased, so as to realize the rapid enhancement of the water evaporation rate. Compared without EC, the water evaporation rate of MWCNTs@HPPS paper with 0.2 mg/100 mL EC solution was increased by 16%. It showed good application potential in the treatment of simulated seawater and dye wastewater with different pH values, which was of great significance for the evaporation process of harsh solutions.

## Figures and Tables

**Figure 1 membranes-12-01208-f001:**
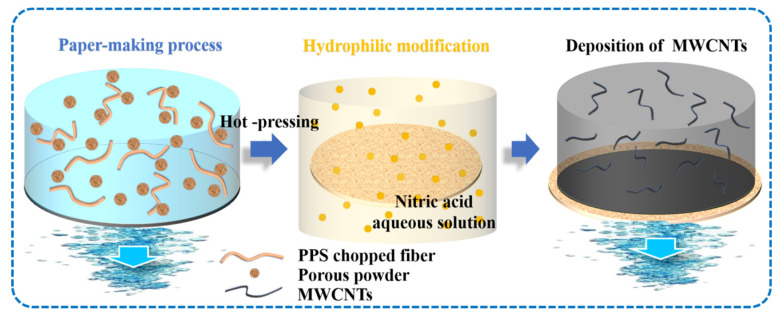
Illustration of the preparation of the MWCNTs@HPPS paper.

**Figure 2 membranes-12-01208-f002:**
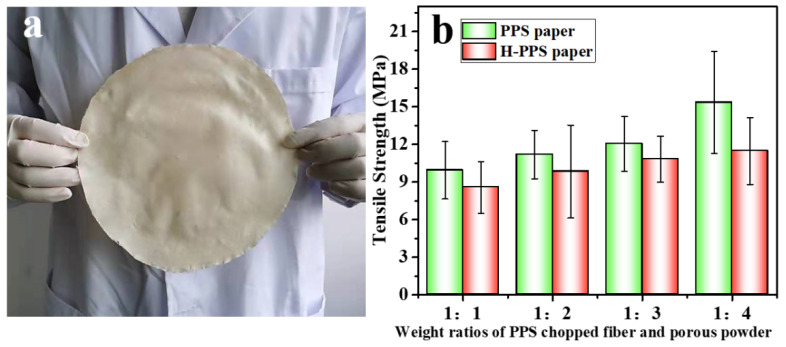
(**a**) Digital images of prepared large-sized PPS paper; (**b**) tensile strength of PPS and HPPS papers prepared with different component contents.

**Figure 3 membranes-12-01208-f003:**
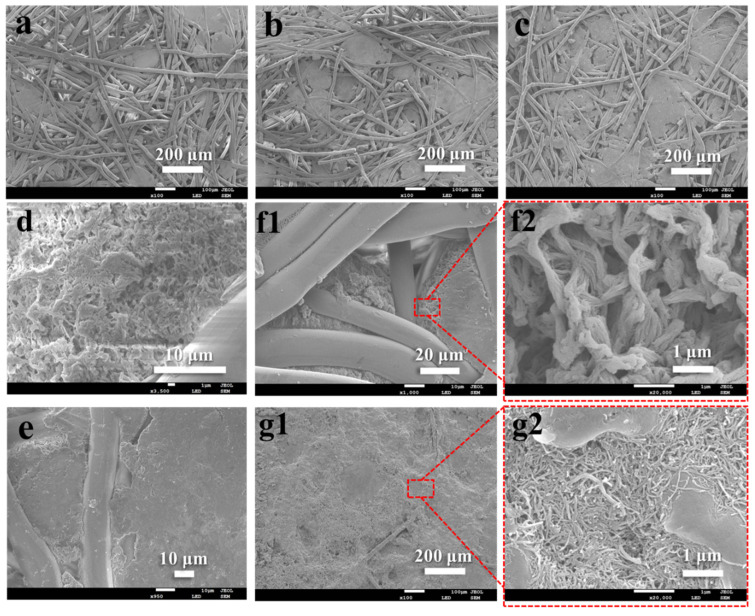
SEM images of PPS papers with weight ratio of PPS chopped fiber and PPS solid solution powder at (**a**) 1:2, (**b**,**d**,**e**,**f**) 1:3, (**c**) 1:4 and (**g**) top surface of MWCNTs@HPPS paper. 1 and 2 represent low and high magnification, respectively.

**Figure 4 membranes-12-01208-f004:**
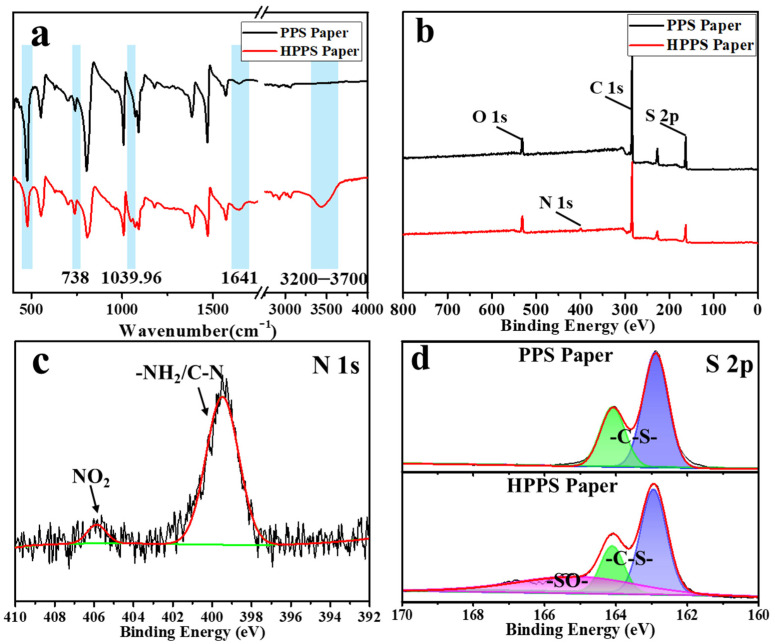
FT-IR spectrum (**a**) and XPS survey spectra (**b**) of PPS and HPPS papers, (**c**) N 1s high-resolution spectra of HPPS paper, (**d**) S 2p high-resolution spectra of PPS and HPPS papers.

**Figure 5 membranes-12-01208-f005:**
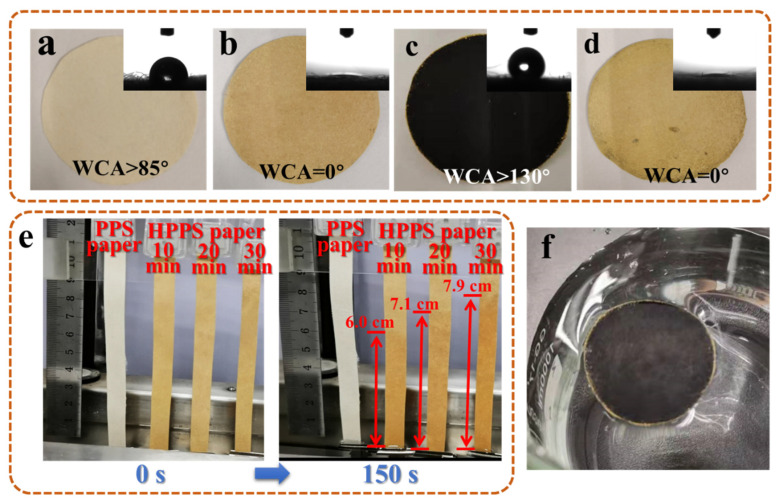
Digital images of prepared PPS paper (**a**), HPPS paper (**b**), and MWCNTs@HPPS paper: top (**c**) and bottom (**d**) surfaces, (**e**) wicking heights after 150 s, (**f**) self-floating ability of the MWCNTs@HPPS paper.

**Figure 6 membranes-12-01208-f006:**
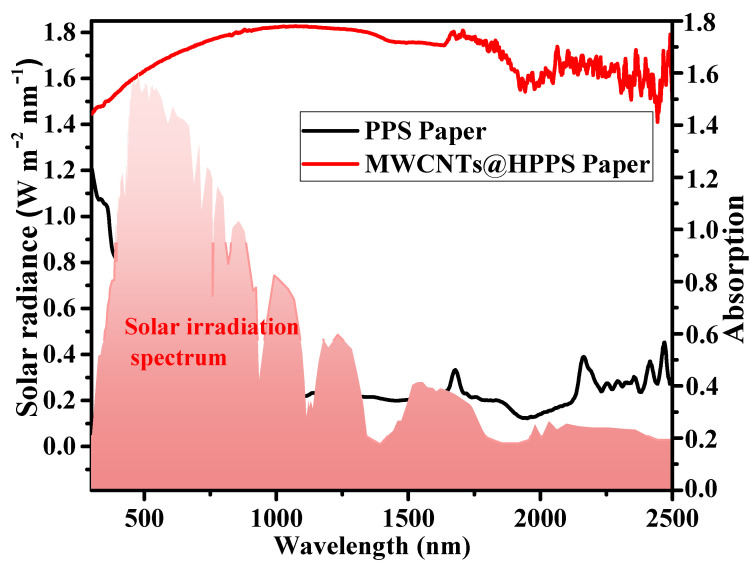
The absorption spectra of PPS and MWCNTs@HPPS papers.

**Figure 7 membranes-12-01208-f007:**
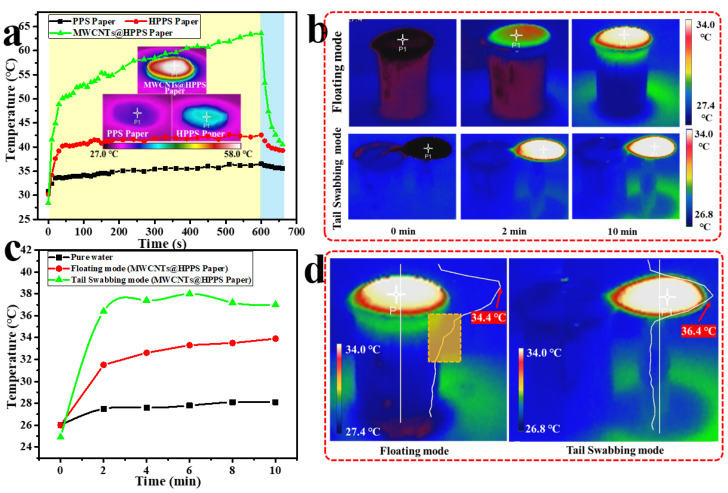
(**a**) Temperature elevation and IR images of the surface of PPS, HPPS and MWCNTs@HPPS papers under 1 sun illumination for 10 min; IR images (**b**) and temperature elevation (**c**) of experimental devices under 1 sun; (**d**) IR images and temperature curve along the axial direction of experimental modes under 1 sun after 10 min.

**Figure 8 membranes-12-01208-f008:**
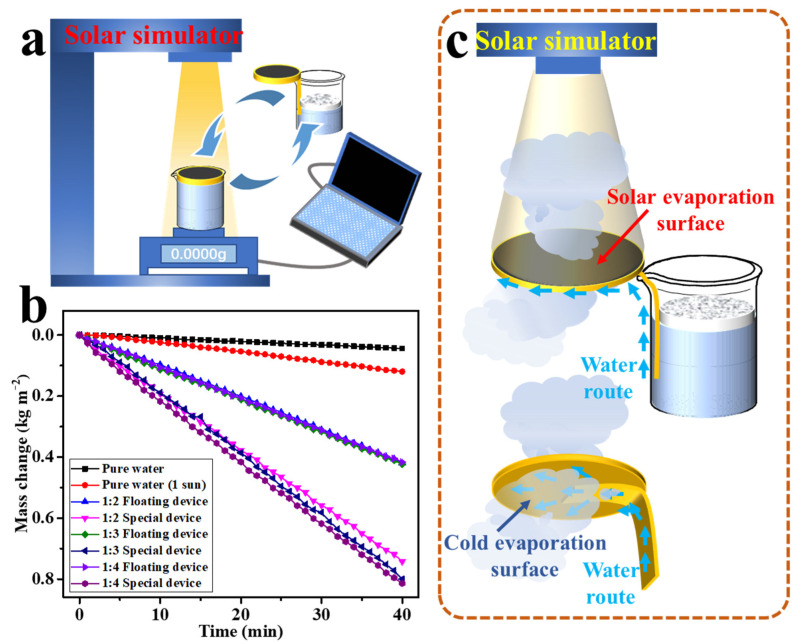
(**a**) Schematic diagram of solar-driven water evaporation experimental device; (**b**) mass changes of water with irradiation time in different evaporation modes under 1 sun; (**c**) schematic diagram of solar-driven interface evaporation process via tail swabbing mode.

**Figure 9 membranes-12-01208-f009:**
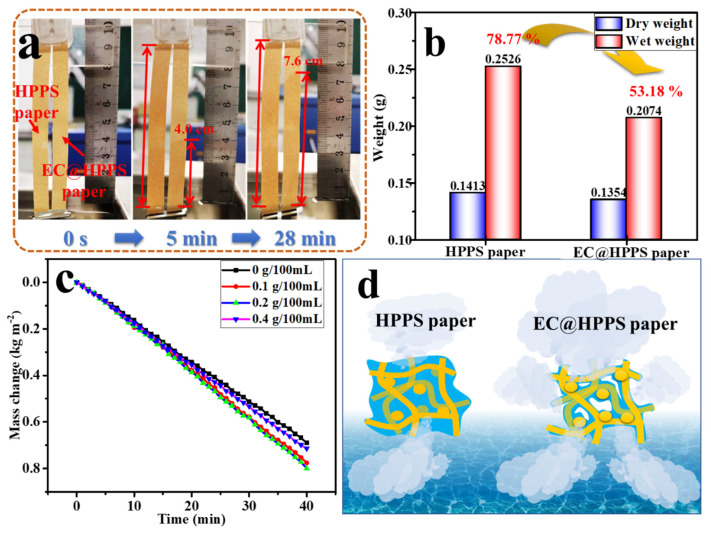
Wicking heights after 5 min and 28 min (**a**) and hydroscopic rate (**b**) of HPPS and EC@HPPS papers; (**c**) water mass changes in light of irradiation time with MWCNTs@HPPS paper prepared with different EC solution under 1 sun; (**d**) schematic diagram of water trace distribution on cold evaporation surface of HPPS and EC@HPPS papers.

**Figure 10 membranes-12-01208-f010:**
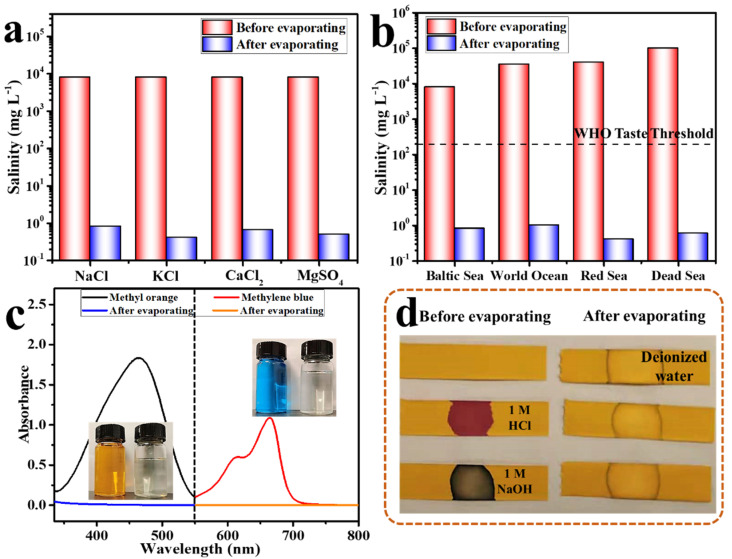
(**a**) Salinities of four simulated seawater samples before and after desalination; (**b**) digital images of MWCNTs@HPPS paper before and after desalination; (**c**) absorbance spectra of dyes solution and the condensed water after purification, the insets showed the corresponding optical images; (**d**) pH value of sewage before and after purification.

## Data Availability

Not applicable.

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
