# Peer review of "Double-Layer MWCNTs@HPPS Photothermal Paper for Water Purification with Strong Acid-Alkali Corrosion Resistance"

_membranes, 2022, doi:10.3390/membranes12121208_

Round 1
Reviewer 1 Report
In this manuscript, the MWCNTs@HPPS paper was fabricated for solar vapor generation. A large-scale photothermal evaporation can be achieved. However, there are some issues in present manuscript. A major revision is needed for improving this work.
1. Some abbreviations have to be clearly explained, eg PPS and HPPS.
2. In the paper, it seems many kinds of materials can work as the photothermal convertor. What is the advantage of the MWCNT?
3. Figure 1 cannot exactly show the preparation procedure of the photothermal paper. What is the porous powder?
4. The properties and performance of the pure fiber and powder (1:0 and 0:1) samples should be given.
5. Figure 3 is not clear. Please do not process SEM images in PowerPoint, especially the scale bar.
6. In Figure 6c, it seems the data plots are sparse. How were such curves obtained?
7. Long-term running performance/stability is missing, eg continuous one-month running.
8. The key parameter of “solar-to-vapor conversion efficiency” is missing.
9. Heat losses of both floating and special devices should be given.
10. The absorption property of the paper is missing.
Author Response
Reviewer #1:
In this manuscript, the MWCNTs@HPPS paper was fabricated for solar vapor generation. A large-scale photothermal evaporation can be achieved. However, there are some issues in present manuscript. A major revision is needed for improving this work.
- Some abbreviations have to be clearly explained, eg PPS and HPPS.
Response: Thanks for your suggestion. We agreed with the reviewer’s comments. We have explained all abbreviations in manuscript-revised.
- In the paper, it seems many kinds of materials can work as the photothermal convertor. What is the advantage of the MWCNT?
Response: As a typical one-dimensional carbon material, MWCNTs have super light absorption and photo-thermal conversion properties, and also have strong hydrophobic properties. They are usually used for hydrophobic surface modification, which is conducive to the preparation of Janus membranes. At the same time, the photo thermal functional layer constructed by MWCNTs can still maintain loose porous structure for water vapor permeation.
- Figure 1 cannot exactly show the preparation procedure of the photothermal paper. What is the porous powder?
Response: Thanks for your suggestion. We have revised Figure 1 in the manuscript, and the porous powder is PPS solid solution powders, which is prepared by thermally induced phase separation (TIPS) method, and the detailed step is performed in Supporting Information.
- The properties and performance of the pure fiber and powder (1:0 and 0:1) samples should be given.
Response: Thanks for your comments. In the paper-making process, both of PPS fiber and powder are indispensable. PPS fiber is the main part of PPS paper, and powder is the adhesive, which plays the role of fixing the fiber. The paper made of pure PPS fiber is loose and can be easily broken, while the paper made of pure powder is a dense film without holes.
- Figure 3 is not clear. Please do not process SEM images in PowerPoint, especially the scale bar.
Response: Thanks for your comments. We have modified Figure 3 and kept the scale bar of the original picture, but the size is too small to be clear. We have also added new scale bar for the convenience of readers.
- In Figure 6c, it seems the data plots are sparse. How were such curves obtained?
Response: Thanks for your comments. This is the temperature curve chart formed according to the IR images (Figure 6b) of the evaporator surface during evaporation. The monitoring frequency is 2 minutes each time.
- Long-term running performance/stability is missing, eg continuous one-month running.
Response: Thanks for your suggestion. We agreed with the reviewer’s comments. And we have added it in manuscript-revised.
In addition, the prepared paper with 0.2 g/100 mL EC solution exhibited excellent long-term evaporation stability for 12 h, where the evaporation rate remained above 1.10 kg·m−2·h−1 (Figure S2).
- The key parameter of “solar-to-vapor conversion efficiency” is missing.
Response: Thanks for your suggestion. We agreed with the reviewer’s comments. And we have added it in manuscript-revised.
- Heat losses of both floating and special devices should be given.
Response: Thanks for your comments. The process of heat loss can be clearly displayed through IR images and temperature curve along the axial direction of experimental modes under 1 sun, and the specific heat loss value will be calculated and analyzed in detail in our next work.
- The absorption property of the paper is missing.
Response: Thanks for your suggestion. We agreed with the reviewer’s comments. We have added the absorption spectra of PPS and MWCNTs@HPPS papers in Figure 6.
MWCNTs were used to construct photothermal conversion layers in this work. As shown in Figure 6, compared with the pristine PPS paper, the prepared MWCNTs@HPPS paper showed high light-absorption.
Reviewer 2 Report
This manuscript demonstrates a MWCNTs@HPPS photothermal paper for solar evaporation. I think some issues have to be addressed before acceptance.
1. In Figure 5c, what is the function of hydrophobic property, especially for the special device? Is it easy to failure?
2. The photos of WAC should be shown.
3. In Figure 9a, the salinity represents the concentration of NaCl or Na+?
4. In Figure 9d, the exact pH values should be given by a pH meter.
5. Please calculate the solar efficiency of the floating and special devices.
6. Please show the optical loss and heat loss of the floating and special devices.
7. Please show the cycle performance of the floating and special devices.
8. The salt-blocking mechanism of the floating and special devices should be explained carefully. (Refs. Nano Energy 2021, 89, 106468; Cell Reports Physical Science 2021, 2, 100310; and J. Mater. Chem. A, 2021, 9, 16233) These review articles systematically summarized all present salt-blocking technologies, and may help the authors deeply analyze the salt-blocking mechanism in this work.
9. The conception of hydrophobic/ hydrophilic double-layer evaporator had been proposed in some typical references (eg Adv. Energy Mater. 2018, 1702884; Mater. Horiz. 2018, 5, 1143-1150; J. Mater. Chem. A, 2018, 6, 16196), which are missing in the Introduction part. These papers started the design of the double-layer evaporator in this research area.
10. Some highly relevant references about MWCNT-based solar evaporator should be mentioned in the Introduction part (eg Adv. Mater. 2020, 1908269; Materials 2022, 15, 929).
Author Response
Reviewer #2:
This manuscript demonstrates a MWCNTs@HPPS photothermal paper for solar evaporation. I think some issues have to be addressed before acceptance.
- In Figure 5c, what is the function of hydrophobic property, especially for the special device? Is it easy to failure?
Response: Thanks for your comments. The hydrophobic surface can prevent salt water or dye wastewater from being transported to the photothermal evaporation layer (MWCNTs layer), and prevent salt from crystallizing in the light absorption layer during water evaporation to affect the light absorption performance.The hydrophobic surface is important for both evaporation modes (floating mode and tail swabbing mode), and its structure is relatively stable.
- The photos of WAC should be shown.
Response: Thanks for your suggestion. We agreed with the reviewer’s comments. And we have added it in manuscript-revised.
- In Figure 9a, the salinity represents the concentration of NaCl or Na+?
Response: Thanks for your comments. The salinity represents the concentration of NaCl.
- In Figure 9d, the exact pH values should be given by a pH meter.
Response: Thanks for your suggestion. We agreed with the reviewer’s comments. And we have added it in manuscript-revised.
- Please calculate the solar efficiency of the floating and special devices.
Response: Thanks for your suggestion. We agreed with the reviewer’s comments. And we have added it in manuscript-revised.
- Please show the optical loss and heat loss of the floating and special devices.
Response: Thanks for your comments. The process of heat loss can be clearly displayed through IR images and temperature curve along the axial direction of experimental modes under 1 sun, and the specific heat loss value will be calculated and analyzed in detail in our next work.
- Please show the cycle performance of the floating and special devices.
Response: Thanks for your suggestion. We agreed with the reviewer’s comments. And we have added it in manuscript-revised.
In addition, the prepared paper with 0.2 g/100 mL EC solution exhibited excellent long-term evaporation stability for 12 h, where the evaporation rate remained above 1.10 kg·m−2·h−1 (Figure S2).
- The salt-blocking mechanism of the floating and special devices should be explained carefully. (Refs. Nano Energy 2021, 89, 106468; Cell Reports Physical Science 2021, 2, 100310; and J. Mater. Chem. A, 2021, 9, 16233) These review articles systematically summarized all present salt-blocking technologies, and may help the authors deeply analyze the salt-blocking mechanism in this work.
Response: Thanks for your suggestion. We have quoted these mentioned literature according to the content of the manuscript.
- The conception of hydrophobic/ hydrophilic double-layer evaporator had been proposed in some typical references (eg Adv. Energy Mater. 2018, 1702884; Mater. Horiz. 2018, 5, 1143-1150; J. Mater. Chem. A, 2018, 6, 16196), which are missing in the Introduction part. These papers started the design of the double-layer evaporator in this research area.
Response: Thanks for your suggestion. We have quoted these mentioned literature according to the content of the manuscript.
- Some highly relevant references about MWCNT-based solar evaporator should be mentioned in the Introduction part (eg Adv. Mater. 2020, 1908269; Materials 2022, 15, 929).
Response: Thanks for your suggestion. We have quoted these mentioned literature according to the content of the manuscript.
==========================================
Further Response:
This manuscript demonstrates a MWCNTs@HPPS photothermal paper for solar evaporation. I think some issues have to be addressed before acceptance.
2. The photos of WAC should be shown.
Response: Thanks for your suggestion. We agree with the reviewer’s comments. And we have added it in manuscript-revised.
4. In Figure 9d, the exact pH values should be given by a pH meter.
Response: Thanks for your suggestion. We think that the pH of the liquid before and after evaporation can be detected more intuitively through the color change of the pH test paper.
6. Please show the optical loss and heat loss of the floating and special devices.
Response: Thanks for your comments. The process of heat loss can be clearly displayed through IR images and temperature curve along the axial direction of experimental modes under 1 sun, which is shown in Fgure 7(c-d). The temperature distribution of bulk water under the two evaporation modes was obvi-ously different, in the floating mode, there was obviously strong heat diffusion to bulk water, which was mainly due to poor insulation performance caused by the loose charac-teristics of PPS paper. In the tail swabbing mode, the MWCNTs@HPPS paper didn’t di-rectly contact bulk water, which greatly limited the heat diffusion, so the temperature change of bulk water was weak and the paper surface temperature was higher than that used in the floating mode. The temperature curve along the axial direction of experimental modes under 1 sun after 10 min were shown in Figure 7d, which further confirmed the above conclusion. And the specific heat loss value will be calculated and analyzed in detail in our next work.
Reviewer 3 Report
1- The manuscript must be improved by checking once again English grammar and technical writing. I have noticed many incomplete and long sentences.
2- The first two sentences in the abstract are too long and not attractive. Revise.
3- The abbreviation “HPPS” in the abstract must be clearly stated.
4- Abbreviations cannot start sentences. Review and address this throughout.
5- The abstract must be checked once again technical writing. Many incomplete and long sentences were noticed.
6- The beginning of the introduction part must be revised to be more attractive.
7- The authors can follow the following references to generally highlight the global contamination and environmental pollution problems at the beginning of the introduction part (https://doi.org/10.3390/catal12050500) ; (https://doi.org/10.3390/ma15134547) (https://doi.org/10.1016/j.jwpe.2022.102847) ; (https://doi.org/10.1016/j.ceramint.2022.05.151) ;. Then, highlight the water desalination technologies.
8- Authors should indicate a clear gap in knowledge which this study seeks to bridge, and potentially contribute to knowledge.
9- Check the spelling error in the caption of Figure 3. All figure captions must be revised to be clearer.
10- Can you give a biotoxicity study?
11- The authors should consider the cost evaluation of using MWCNTs@HPPS Janus paper in large scale applications.
12- The authors should identify the limitations of this study and the recommended future studies in the conclusion part.
13- The reference style must be unified.
Author Response
Reviewer #3:
- The manuscript must be improved by checking once again English grammar and technical writing. I have noticed many incomplete and long sentences.
Response: Thanks for your suggestion. And we have reread the entire manuscript and revised the manuscript.
- The first two sentences in the abstract are too long and not attractive.
Response: Thanks for your suggestion. We agreed with the reviewer’s comments. And we have revised it in manuscript-revised.
- The abbreviation “HPPS” in the abstract must be clearly stated.
Response: Thanks for your suggestion. We agreed with the reviewer’s comments. We have explained it in manuscript-revised.
- Abbreviations cannot start sentences. Review and address this throughout.
Response: Thanks for your suggestion. We agreed with the reviewer’s comments. And we have revised it in manuscript-revised.
- The abstract must be checked once again technical writing. Many incomplete and long sentences were noticed.
Response: Thanks for your suggestion. We agreed with the reviewer’s comments. And we have revised it in manuscript-revised.
- The beginning of the introduction part must be revised to be more attractive.
Response: Thanks for your suggestion. We agreed with the reviewer’s comments. And we have revised it in manuscript-revised.
- The authors can follow the following references to generally highlight the global contamination and environmental pollution problems at the beginning of the introduction part.
(https://doi.org/10.3390/catal12050500);
(https://doi.org/10.3390/ma15134547);
(https://doi.org/10.1016/j.jwpe.2022.102847); (https://doi.org/10.1016/j.ceramint.2022.05.151).
Then, highlight the water desalination technologies.
Response: Thanks for your suggestion. We have quoted this mentioned literature according to the content of the manuscript.
- Authors should indicate a clear gap in knowledge which this study seeks to bridge, and potentially contribute to knowledge.
Response: Thanks for your suggestion. We agreed with the reviewer’s comments. And we have added it in the conclusion part.
In conclusion, based on mature papermaking technology and hot-pressing process, we have developed a double-layer MWCNTs@HPPS Janus paper for solar driven interface evaporation. Especially under the tail swabbing mode, the water evaporation rate of the prepared paper reached 1.23 L/m2·h, which was mainly due to interfacial evaporation and cold surface evaporation. By adjusting the water absorption of the cold evaporation surface, the evaporation area of the cold evaporation surface could be significantly increased, so as to realize the rapid enhancement of the water evaporation rate. Compared without EC, the water evaporation rate of MWCNTs@HPPS paper with 0.2 mg/100mL EC solution was increased by 16 %. It showed good application potential in the treatment of simulated seawater and dye wastewater with different pH values, which was of great significance for the evaporation process of harsh solutions.
- Check the spelling error in the caption of Figure 3. All figure captions must be revised to be clearer.
Response: This is the author's writing error, and we have revised the entire manuscript.
- Can you give a biotoxicity study?
Response: Thanks for your suggestion. Biotoxicity, that is, antibacterial property, is a very important property for water treatment materials. This work focuses on the analysis of the influence of structural design on the evaporation rate. In the next work, we will consider testing its antibacterial property.
- The authors should consider the cost evaluation of using MWCNTs@HPPS Janus paper in large scale applications.
Response: Thanks for your suggestion. Based on the mature papermaking process, we propose the possibility of realizing large-scale production, which is of great significance for practical application. In the next work, we will focus on the price of raw materials and the simplification of processes to reduce costs, and conduct corresponding cost accounting.
- The authors should identify the limitations of this study and the recommended future studies in the conclusion part.
Response: Thanks for your suggestion. We agreed with the reviewer’s comments. And we have added it in the conclusion part.
It showed good application potential in the treatment of simulated seawater and dye wastewater with different pH values, which was of great significance for the evaporation process of harsh solutions.
- The reference style must be unified.
Response: Thanks for your suggestion. We agreed with the reviewer’s comments. And we have revised it in manuscript-revised.
Round 2
Reviewer 1 Report
I think the present version can be accepted.
Author Response
Thank you for your approval.
Reviewer 2 Report
I did not see the revision of Comment 2, 4, and 6 in the manuscript as the authors replied.
Author Response
- The photos of WAC should be shown.
Response: Thanks for your suggestion. We agree with the reviewer’s comments. And we have added it in manuscript-revised.
- In Figure 9d, the exact pH values should be given by a pH meter.
Response: Thanks for your suggestion. We think that the pH of the liquid before and after evaporation can be detected more intuitively through the color change of the pH test paper.
- Please show the optical loss and heat loss of the floating and special devices.
Response: Thanks for your comments. The process of heat loss can be clearly displayed through IR images and temperature curve along the axial direction of experimental modes under 1 sun, which is shown in Figure 7(c-d). The temperature distribution of bulk water under the two evaporation modes was obviously different, in the floating mode, there was obviously strong heat diffusion to bulk water, which was mainly due to poor insulation performance caused by the loose characteristics of PPS paper. In the tail swabbing mode, the MWCNTs@HPPS paper didn’t directly contact bulk water, which greatly limited the heat diffusion, so the temperature change of bulk water was weak and the paper surface temperature was higher than that used in the floating mode. The temperature curve along the axial direction of experimental modes under 1 sun after 10 min were shown in Figure 7d, which further confirmed the above conclusion. And the specific heat loss value will be calculated and analyzed in detail in our next work.

Reviewer 3 Report
Accept.
Author Response
Thank you for your approval.